# Obesity Augments Glucocorticoid-Dependent Muscle Atrophy in Male C57BL/6J Mice

**DOI:** 10.3390/biomedicines8100420

**Published:** 2020-10-15

**Authors:** Laura C. Gunder, Innocence Harvey, JeAnna R. Redd, Carol S. Davis, Ayat AL-Tamimi, Susan V. Brooks, Dave Bridges

**Affiliations:** 1Department of Nutritional Sciences, University of Michigan School of Public Health, Ann Arbor, MI 48109, USA; lcgunder@umich.edu (L.C.G.); iharvey@umich.edu (I.H.); reddj@umich.edu (J.R.R.); ayata@umich.edu (A.A.-T.); 2Department of Physiology, University of Tennessee Health Science Center, Memphis, TN 38103, USA; 3Adipocyte Biology Laboratory, Pennington Biomedical Research Center, Louisiana State University, Baton Rouge, LA 70803, USA; 4Department of Molecular & Integrative Physiology, University of Michigan Medical School, Ann Arbor, MI 48109, USA; csdav@umich.edu (C.S.D.); svbrooks@umich.edu (S.V.B.); 5Department of Pediatrics, University of Tennessee Health Science Center, Memphis, TN 38103, USA

**Keywords:** glucocorticoids, atrophy, obesity, atrogenes, insulin resistance

## Abstract

Glucocorticoids promote muscle atrophy by inducing a class of proteins called atrogenes, resulting in reductions in muscle size and strength. In this work, we evaluated whether a mouse model with pre-existing diet-induced obesity had altered glucocorticoid responsiveness. We observed that all animals treated with the synthetic glucocorticoid dexamethasone had reduced strength, but that obesity exacerbated this effect. These changes were concordant with more pronounced reductions in muscle size, particularly in Type II muscle fibers, and potentiated induction of atrogene expression in the obese mice relative to lean mice. Furthermore, we show that the reductions in lean mass do not fully account for the dexamethasone-induced insulin resistance observed in these mice. Together, these data suggest that obesity potentiates glucocorticoid-induced muscle atrophy.

## 1. Introduction

Skeletal muscle is vital to normal functioning and the maintenance of health. Muscle is critical to the regulation of lipid, glucose and amino acid metabolism, processes which are commonly dysregulated in association with illness [1]. Many factors including age, poor nutrition, lack of exercise, medication, stress and disease can lead to loss of skeletal muscle mass and function, with attendant reductions in lifespan and health span [2]. One causal factor in muscle loss is elevated glucocorticoids, either pharmacologically or as the result of chronic stress. It is estimated that 1–2% of individuals in the USA and UK are prescribed glucocorticoids [3,4]. Similarly, chronically elevated glucocorticoids are associated with higher longitudinal risk of metabolic diseases [5,6]. While obesity affects approximately 40% of the American population [7], increasing the risk of type 2 diabetes, cardiovascular diseases and liver disease among other comorbidities [8], the combination of glucocorticoids and obesity on outcomes of metabolic health has received little attention. 

Elevated levels of glucocorticoids have been shown to cause skeletal muscle atrophy and weakness [9,10,11,12,13]. This muscle atrophy is due to increased muscle proteolysis and inhibition of protein synthesis [12,14], linked mechanistically to an upregulation of atrogenes (a class of E3 ubiquitin ligases) and a downregulation of mTORC1, as well as other factors [12,15,16,17,18,19,20]. Previous work by our group and others has demonstrated that glucocorticoids and obesity may have synergistically detrimental effects [14,21,22,23,24].

In this manuscript, we provide data indicating that both lean and obese mice have reductions in lean mass, muscle mass, and strength when treated with dexamethasone, and that these effects are worsened in obese mice. We show that obese dexamethasone-treated mice have elevated inductions of key atrophy-inducing transcripts including *Fbxo32* and *Trim63* (encoding Atrogin-1 and MuRF1, respectively) and their upstream regulator *Foxo3*. Lastly, we show that obese dexamethasone-treated mice are profoundly insulin-resistant, even after accounting for reduced muscle mass.

## 2. Results

In order to assess diet-induced obesity in mice, we randomized mice into diets of normal chow (NCD) or high fat diet (HFD), then after 12 weeks on their respective diets randomized again into treatment groups (dexamethasone or water). Prior to randomization into dexamethasone treatments, high fat diet animals had approximately the same percent body fat mass of 30%. Upon randomization, we evaluated food intake during the course of treatment to determine the possible origin of changes in adiposity. HFD–dexamethasone animals consumed approximately 70% more calories per day than water controls. Even though the HFD–dexamethasone mice ate the most calories, they lost both fat and lean mass when compared to their HFD–water counterparts (Table 1), consistent with our prior data [21]. This is suggestive of either increased energy expenditure or decreased digestive efficiency in these animals.

Our prior work demonstrated substantial elevations of dexamethasone ingestion over a five-week period in obese mice, an effect we proposed was secondary to their diabetic phenotype [21]. In this shorter exposure, while we noted a 36% reduction in fluid intake in both groups of dexamethasone-treated mice, there was no moderating effect of HFD treatment (*p* = 0.85; Table 1) indicating equivalent dexamethasone doses between NCD and HFD mice.

### 2.1. Greater Losses in Grip Strength in Obese Dexamethasone-Treated Mice

To assess the effect of glucocorticoids on overall muscle strength, we measured grip strength. Dexamethasone treatment resulted in reductions in grip strength in both lean and obese mice when compared to their non-treated counterparts (Figure 1A,B). Obese dexamethasone-treated mice had greater overall losses in grip strength when compared to the lean animals. We observed a 4.8% reduction in grip strength for lean animals (*p* = 0.007) but a 26.2% reduction in obese animals (*p* = 3.6 × 10^−5^).

### 2.2. Reductions in Strength are Related to Smaller Cross-Sectional Area

In order to expand upon these results, we measured the force generated by gastrocnemius muscle *in situ*. These experiments were performed by stimulation the tibial nerve and by direct electrical stimulation of the muscle. In NCD animals, the force generated by nerve stimulation was reduced 10% when treated with dexamethasone. However, in HFD animals, force generated by nerve stimulation was reduced 32% in animals treated with dexamethasone, with a significant interaction between pre-existing obesity and dexamethasone treatment (*p*_interaction_ = 0.009, Figure 1C). Similarly, in NCD animals, force generated by direct muscle stimulation was reduced 11% when treated with dexamethasone, while in HFD animals, the force generated by direct muscle stimulation was reduced 30% when treated with dexamethasone relative to control animals (*p*_interaction_ = 0.024, Figure 1D). Dexamethasone had significant effects in both groups for both muscle (*p* = 0.016 for NCD and *p* = 0.005 for HFD via Student’s *t*-tests) and nerve stimulation (*p* = 0.015 for NCD and *p* = 0.003 for HFD). These data suggest primarily a muscle-autonomous phenotype rather than the presence of functional denervation, as the weakness was comparable with nerve and direct muscle stimulation. This also suggests that hyperglycemia-induced peripheral neuropathy is not a major explanation for these reductions.

In order to examine whether changes in muscle strength were proportional to declines in muscle size, we plotted a regression of force versus whole-muscle cross-sectional area (CSA; Figure 1E,F). The quadriceps CSA was significantly lower for the dexamethasone-treated groups, and this was enhanced by obesity (Figure 2C). The CSA explained 64% and 59% of the variance in force stimulated at the nerve and muscle, respectively. As the cross-sectional area declined, muscle force by both stimulations decreased in proportion. Regression modeling showed that pre-existing obesity did not significantly modify this force–CSA relationship (nerve stimulation: *p* = 0.47; muscle stimulation: *p* = 0.42). These data indicate that pre-existing obesity causes elevated dexamethasone-induced muscle weakness, but that this is largely explained by reductions in muscle size rather than qualitative defects in the force-generating machinery within the muscle.

### 2.3. Enhanced Muscle Atrophy in Obese Mice

The obese dexamethasone-treated animals had larger reductions in fat free mass (Figure 2A), gastrocnemius weight, and whole-muscle cross-sectional area (Figure 2B,C) when compared to their lean counterparts. At sacrifice, the NCD animals’ gastrocnemius weights were smaller after treatment with dexamethasone by 13% in the NCD treated group, but by 27% in the HFD group (*p*_interaction_ = 0.021). Similarly, the cross-sectional area of the muscle was reduced 13% in the NCD group and 23% in the HFD group, though the modifying effect of obesity did not quite reach statistical significance (*p*_interaction_ = 0.11). There were no significant changes in relative gastrocnemius weights due to dexamethasone treatment after normalization to total body weight (*p*_interaction_ = 0.486, *p*_dexamethasone_ = 0.386). We interpret these data to indicate that the individual muscle mass changes were proportional to changes in total body weight, and that this was largely driven by reductions in lean mass.

### 2.4. Obesity with Dexamethasone Treatment Resulted in Smaller Type II Muscle Fibers

In order to assess changes at the individual muscle fiber level, we cryosectioned the quadriceps from mice at the mid-belly and H&E stained these samples (Figure 2D). The lean animals’ muscle fibers were reduced by 17% in the dexamethasone treated groups, while obese animals’ muscle fibers were reduced by 55% in the dexamethasone treated mice (*p*_interaction_ = 0.001; Figure 2E).

In order to evaluate any changes in the ratio of oxidative versus non-oxidative fiber types, we stained muscle sections and quantified the muscle fibers based upon their oxidative capacity. We used NADH/NBT staining, which is responsive to succinate dehydrogenase activity. Mouse skeletal muscle is made up Type I, Type IIa, Type IIb, and Type IIx fibers [25,26]. Oxidative fibers or Type I fibers stain the darkest (Figure 2F). We found no significant change in the ratio of oxidative to total fibers in the mice quadriceps in lean or obese mice treated with dexamethasone (Figure 2G).

Using these cryosections, we next tested for fiber-type specific reductions in fiber size. There was a main effect of dexamethasone treatment in all fiber types except oxidative (*p* = 0.001 for light-, *p* = 0.004 for medium- and *p* = 0.449 for dark-stained fibers). There was a significant main effect of diet reducing fiber size in light- (*p* = 0.01) but not medium- (*p* = 0.125) or dark-stained fiber (*p* = 0.425). We found that dexamethasone treatment reduced Type IIa or light-stained fibers’ CSA in lean and obese mice by 28% and 40%, respectively, though the moderating effect of obesity did not reach statistical reference (*p*_interaction_ = 0.49). Dexamethasone treatment also reduced Type IIb or medium-stained fibers’ CSA in lean and obese mice by 35% and 32%, respectively (*p*_interaction_ = 0.58). As for Type I or dark-stained fibers, dexamethasone treatment only reduced fibers’ CSA in NCD animals. Dexamethasone treatment reduced Type I fibers’ CSA by 21% in lean, the treatment increased fibers’ CSA in obese mice by 14% (*p*_interaction_ = 0.0031; Figure 2H).

### 2.5. Obesity and Dexamethasone Cause Elevated Atrogene Expression

To evaluate the molecular effects of dexamethasone in vivo and how this was moderated by obesity, we determined atrogene expression in quadriceps after a two-week treatment time course, with animals euthanized at 0, 3, 7, and 14 days. After one week of dexamethasone treatment, we observed induction of *Foxo3* and the atrogenes, *Trim63* (Atrogin-1) and *Fbxo32* (MuRF1), to be greater in obese mice compared to their lean counterparts, though the interaction between obesity status and dexamethasone treatment did not reach statistical significance for these transcripts (Figure 3). We did not observe a treatment effect in either diet for *Foxo1* or *Ncr31* (the gene that encodes for the glucocorticoid receptor). These data suggest that the obesity-sensitizing effects on muscle atrophy could be related to transcriptional elevations of *Foxo3* and these two atrogenes.

### 2.6. Obese Dexamethasone-Treated Mice Are Insulin Resistant After Adjusting for Muscle Mass

We evaluated insulin sensitivity in these mice, as the majority of all postprandial glucose uptake occurs within the muscle [27]. In lean animals, there was no significant change in fasting blood glucose following dexamethasone treatment; however, there was a 44% increase in fasting blood glucose in obese animals given dexamethasone (*p*_interaction_ = 0.033; Figure 4A), which is consistent with our previous report [21]. In order to evaluate whether the dexamethasone-treated animals were insulin resistant beyond what was expected by reductions in muscle mass, we adjusted insulin concentrations according to lean mass. In both lean and obese animals, dexamethasone induced near complete insulin resistance (p = 8.8 × 10^−12^ for NCD and 7.7 × 10^−7^ for HFD; Figure 4B). These data suggest that even after accounting for change in muscle mass, glucocorticoids still cause insulin resistance. To test whether proximal insulin signaling was affected in either group, we evaluated muscle lysates from gastrocnemius tissues at the end of a hyperinsulinemic euglycemic clamp. We found that the relative phosphorylation of Akt at Ser 473 did not change between water and dexamethasone treatments in either group (Figure 4C,D). This is consistent with prior work demonstrating that proximal insulin signaling is largely unaffected by glucocorticoids [28,29].

## 3. Discussion

Here we demonstrate that dexamethasone treatment in concert with pre-existing obesity causes pronounced reductions in muscle strength and size, and causes insulin resistance in mice. In our previous study we demonstrated via euglycemic hyperinsulinemic clamps that obese dexamethasone-treated mice were insulin resistant (as determined by suppressed glucose infusion rates), had lower muscle glucose uptake, and had elevated endogenous glucose production. Based on elevated lipolysis in these mice, we posited that this is due to indirect promotion of glucose production by adipocyte lipolysis [21].

Muscle weakness is a common side effect of elevated exogenous and endogenous glucocorticoids [12,30]. For example, adults who have elevated salivary cortisol have a significantly higher risk of loss of grip strength than their peers [11]. This work could be particularly important because those with obesity are more likely to have reduced muscle function [31,32,33,34]. Notably, people with obesity are also more likely to have elevations in endogenous glucocorticoid levels [35,36]. Our model used exogenous glucocorticoid treatment in the form of dexamethasone, a fluorinated synthetic glucocorticoid with high selectivity for the glucocorticoid receptor (GR) over the mineralocorticoid receptor. Our dose of dexamethasone treatment is equivalent to a human dose of 80 μg/kg/d, which is comparable to a high therapeutic dose administered to human patients, with a usual range from 2–200 μg/kg/d [37,38,39,40].

Glucocorticoids induce muscle atrophy in a muscle and fiber-type specific manner. Specifically, and consistent with our findings, Type II fibers are more prone to glucocorticoid-induced changes in cross-sectional areas [9,10,12,30,41]. It is plausible that a selective loss in non-oxidative fiber functionality could reduce a human’s ability to use short bursts of energy, make rapid postural changes, or lift heavy objects [42]. The mechanisms causing differential specificity to glucocorticoids between fibers are not clear, to our knowledge.

The mechanisms underlying how increased responsiveness to dexamethasone in obese animals occurs are not currently understood. Our data are concordant with a report showing that glucocorticoids given simultaneously with HFD enhances muscle decay and exacerbates induction of atrogenes [24]. We did not observe transcriptional increases in GR in muscle (Figure 3) or adipose tissue [21] in obese animals that would explain these findings. One hypothesis is that obesity remodels the chromatin landscape, allowing for easier GR access to genes involved in modulating muscle size and function. Indeed, obesity alters the packing and accessibility of DNA in adipocytes [14,21,43], and therefore may have a similar effect in muscle in which glucocorticoid response elements are more easily bound by GR, causing increased glucocorticoid action. Another potential mechanism is that the effects of GR-dependent signaling are enhanced by insulin resistance as a result of FOXO dephosphorylation, though in our case we observed substantial transcriptional activation of *FOXO3*. Another possibility is that reduced muscle mass results in limited physical activity, which in turn reduces insulin sensitivity. In this work, we observed transcriptional upregulation of the genes mentioned above, but have not assessed activities or levels of these GR inducible proteins.

Glucocorticoids and obesity both have deleterious health effects. These effects include loss of skeletal muscle, which may result in reduced motor function, coordination, and energy production [31,33,34]. Insulin resistance is an additional negative effect associated with both elevated glucocorticoids and excess adiposity in the body [15,21,44]. The process by which these factors induce insulin resistance is not yet fully understood. In this study, we highlight that dexamethasone-induced muscle atrophy was exacerbated in an obese mouse model, as evidenced by synergistic reductions in muscle function, muscle mass, and fiber-specific cross-sectional areas. Based on this and prior findings that showed dexamethasone treatment exacerbates insulin resistance and NAFLD in the context of obesity, it should be considered whether humans with obesity are more prone to stress or drug-induced glucocorticoid responses when prescribing steroids.

## 4. Materials and Methods

### 4.1. Animal Husbandry

Male C57BL/6J mice were purchased from the Jackson Laboratory at nine weeks of age and randomized into groups of three to four animals per cage. All animals were on a light/dark cycle of 12 hours and housed at 22 °C. At 10 weeks of age, mice were placed on a high fat diet (HFD; 45% fat from lard, 35% carbohydrate mix of starch, maltodextrin, and sucrose, and 20% protein from casein, Research Diets cat no. D12451) or kept on a normal chow diet (NCD; 13% fat, 57% carbohydrate, and 30% protein; Teklad catalog no. 5LOD) for 12 weeks. At 22 weeks, mice were either treated with vehicle (water) or approximately 1 mg/kg/d of water-soluble dexamethasone (Sigma-Aldrich, St. Louis, MO, USA; catalog no. D2915) dissolved in their drinking water. All mice were provided with ad libitum access to food and their respective waters throughout the study. Food and liquid consumption were measured weekly to determine the concentration of dexamethasone consumed per cage and volumes were averaged per mouse per cage. All animal procedures were approved by the University of Michigan or University of Tennessee Health Sciences Center Institutional Animal Use and Care Committees.

### 4.2. Grip Strength

Mice were tested using a grip strength meter with a Chatillon digital force gauge (AMETEK, Berwyn, PA, USA). Mice were placed on a grid attached to the meter and once all four paws had contact with the grid, the mice were slowly pulled backwards by the tail until they left the grid. Each mouse was tested five times and given approximately 10 seconds rest between each test. Final measurements for grip strength were assessed by taking the average of the five trials and reported as average peak force (N).

### 4.3. In situ Contractile Measurements

After isoflurane-induced anesthesia, the right gastrocnemius muscle was carefully isolated and a 4–0 silk suture was tied around the distal tendon. After the tendon was secured, it was cut so the hindlimb could be secured at the knee to a fixed post. Animals were placed on a temperature-controlled platform with a continuous drip of saline over the muscle at 37 °C to keep it warm and hydrated. The distal tendon of the gastrocnemius muscle was tied to the lever arm of a servomotor (6650LR, Cambridge Technology, Cambridge, MA, USA). In order to measure the force generated under circumstances where the neuromuscular transmission of action potentials activates muscle fibers, a bipolar platinum wire electrode was used to stimulate the tibial nerve. The voltage of the electrode pulses was incrementally adjusted to find the maximum isometric twitch, and the muscle length was altered to find the optimal length (L_o_). Optimal length is the length of the muscle in which the maximal twitch force was obtained. Once L_o_ was found, gastrocnemius muscles were kept at that length (L_o_) and the frequency of 300-ms trains of pulses was increased in increments of 50 Hz until maximum isometric tetanic force (P_o_) was achieved. Muscles were rested for one minute in between stimulations. In order to measure the force generated in response to direct depolarization of the muscle fibers bypassing the requirement for neuromuscular transmission of the activating stimulus, an electrode cuff was placed around the mid-belly of gastrocnemius for muscle stimulation. The same process was then repeated as described above for nerve-stimulated contractions. After all force measurements, mice were sacrificed and both gastrocnemius and quadricep muscles were dissected, weighed, and snap frozen in liquid nitrogen and stored at −80 °C.

### 4.4. Histology and Fiber Type Quantifications

Quadriceps were collected and frozen in 2-methyl-butane cooled under liquid nitrogen. Quadricep samples were sectioned using a CryoStar NX350 HOVP Cryostat (Thermo Scientific, Waltham, MA, USA) at −20 °C with a thickness of 10 μm through the mid-belly and mounted on SuperFrost glass slides (Electron Microscopy Sciences, Hatfield, PA, USA, catalog no. 71882-01). For analysis of fiber cross-sectional areas (CSAs), fibers were assessed by hematoxylin and eosin (H&E staining), while for fiber type, muscles were stained using NADH/NBT (NADH and nitro blue tetrazolium) staining as described in [25,45]. Light-stained fibers were labeled as Type IIb fibers, medium-stained fibers as Type IIa, and dark-stained as Type I fibers. The images were taken using a 20× objective of an EVOS XL digital inverted microscope (Life Technologies, Carlsbad, CA, USA). Muscle fibers were individually counted in each image by a blinded investigator, and the cross-sectional areas were measured by outlining 150 randomly chosen fibers per image and using ImageJ [46]. 

### 4.5. mRNA Quantification

Cells and tissues were lysed in TRIzol using a TissueLyser II (Qiagen, Hilden, Germany), and RNA was extracted using a PureLink RNA kit (catalog no. 12183025; Life Technologies) following the manufacturer’s instructions. Complementary DNA (cDNA) was synthesized using the High Capacity cDNA Reverse Transcription Kit without an RNAse inhibitor (catalog no. 4368813; Life Technologies). Quantitative real-time polymerase chain reaction (qPCR) was performed using a QuantStudio 5 (Thermo Fisher Scientific) with primers, cDNA, and Power SYBR Green PCR Master Mix (catalog no. 4368708; Life Technologies) per the manufacturer’s instructions. Messenger RNA (mRNA) expression levels were normalized to a control gene, *Pgk1*, after evaluating eight control gene candidates (primer sequences in Table 2).

### 4.6. Protein Quantification

Quadriceps were dissected from mice after two hours of insulin infusion under euglycemic conditions. Glucose clamped conditions are described in [47]. Muscles were lysed in RIPA buffer using a TissueLyser II, followed by centrifugation at 20,000× *g*. Lysates were combined with reducing agent and SDS loading buffer and loaded onto SDS-PAGE gradient gels, then transferred to nitrocellulose membranes. Membranes were blotted using pAkt Ser 473 (cat no. 4060; RRID AB_2314509) and Akt (cat no. 9272; RRID AB_329827) antibodies from Cell Signaling Technologies. Blots were scanned and quantified using a LiCOR imaging system.

### 4.7. Assessment of Insulin Tolerance

Insulin tolerance testing took place between ZT8 and ZT10 following a six-hour fast. Mice were assessed for glucose levels using a handheld glucometer (Accuchek) with blood drawn from the tail vein. Insulin (Humulin R, Lilly) was then administered via intraperitoneal injection at 0.75 IU per kg of lean mass for lean mice (determined by echo MRI) and 1.5 IU per kg of lean mass for obese mice. Different insulin doses were used to obtain similar glucose responses in control mice. Glucose was measured in 15 minutes intervals for a total of two hours following insulin administration. 

### 4.8. Body, Fat, and Lean Mass Determination

Body weight was measured using a digital scale. Fat and lean mass were determined using an EchoMRI 2100 (EchoMRI), without sedation or anesthesia. 

## 5. Statistics

All results are represented as mean ± SEM. Two-way ANOVA analyses, mixed linear models and chi-squared tests were performed to test for significance and determine interactions between diet and dexamethasone treatment. Pairwise testing was performed after assessing normality and equality of variances. If a Shapiro–Wilk test was insignificant, a Levene’s test was performed, followed by a Welch’s or Student’s *t*-test as noted in the text. For non-normally distributed data, a Mann–Whitney U-test was used. A *p*-value under 0.05 was considered significant. All statistical tests were conducted using R version 3.5.0 [48]. All raw data and analysis scripts are available at http://bridgeslab.github.io/CushingAcromegalyStudy/.

## Figures and Tables

**Figure 1 biomedicines-08-00420-f001:**
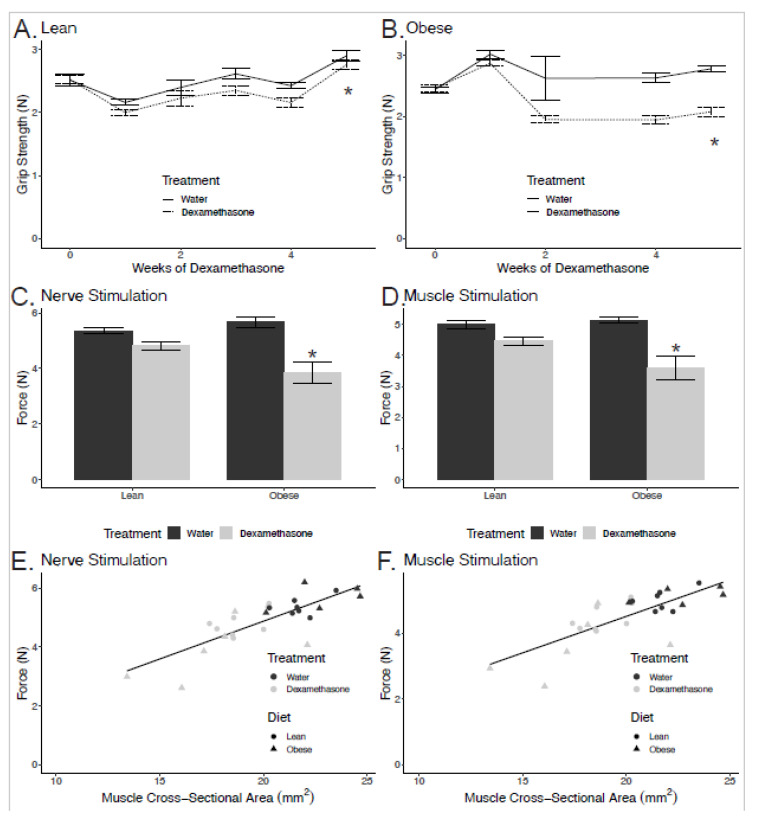
Obese dexamethasone-treated mice have reduced muscle strength. Grip strength in lean (**A**) and obese (**B**) male mice over five weeks of dexamethasone treatment (n = 4–8 per group). Mice were fed a high fat diet (HFD) or normal chow diet (NCD) for 12 weeks prior to randomization into water or dexamethasone treatments. Force generated by nerve stimulation (**C**) and direct muscle stimulation (**D**) of the gastrocnemius muscle in lean and obese mice treated with vehicle (water) or dexamethasone for 15–21 days after 12 weeks of NCD or HFD. Force plotted relative to whole muscle cross-sectional area (CSA) (**E**,**F**). Asterisks indicate significant interaction between diet and treatment by two-way ANOVA, except for panels (**A**,**B**), where they indicate a difference between treatments (n = 5–8 per group).

**Figure 2 biomedicines-08-00420-f002:**
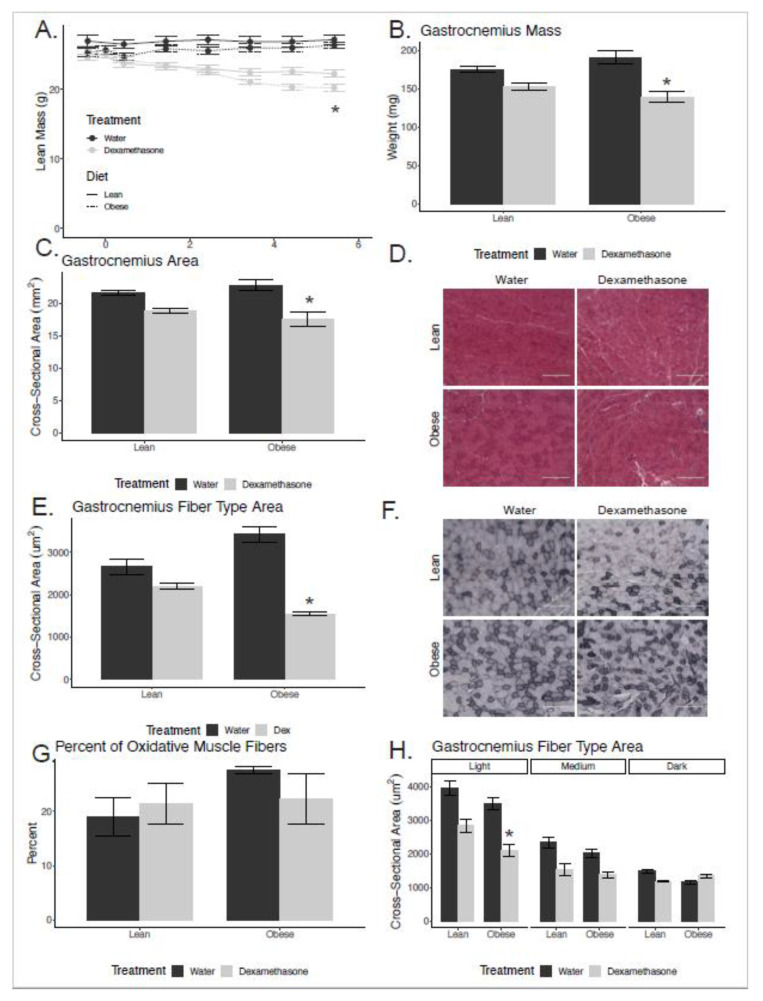
Obese dexamethasone-treated mice have reduced muscle size. (**A**) Lean mass determined via EchoMRI. Gastrocnemius muscle mass (**B**) and cross-sectional area ((**C**) from lean and obese mice treated with vehicle (water) or dexamethasone (n = 5–8 per group). H&E-stained section of muscles (quadriceps; (**D**)). Average fiber cross-sectional area (**E**) averaged from 200 fibers per section (quadriceps; n = four mice per group). NADH/NBT-stained section of muscles (quadriceps; (**F**)) from mice treated with water or dexamethasone for six weeks. Percent of slow-oxidative or Type I fibers to total fibers ((**G**); n = four sections per group). Average fiber cross-sectional area separated by NADH/NBT staining density with dark fibers indicating slow-oxidative or Type I muscle fibers (quadriceps muscle; (**H**)). Asterisks indicate a significant interaction between diet and dexamethasone treatment by two-way ANOVA.

**Figure 3 biomedicines-08-00420-f003:**
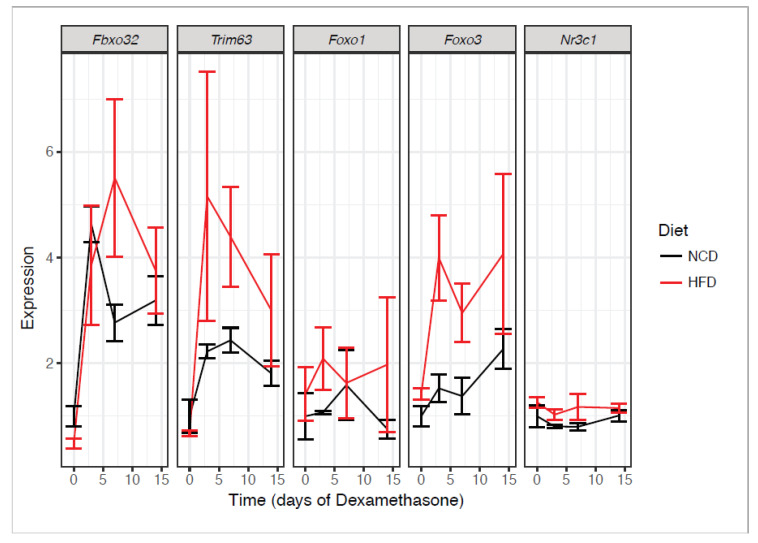
Obesity enhances dexamethasone-induced muscle degradation transcripts. Atrogene expression in NCD or HFD mice treated with dexamethasone for the indicated time points and euthanized ad libitum. mRNA was extracted and quantified from quadriceps muscles. n = 6–8 per group.

**Figure 4 biomedicines-08-00420-f004:**
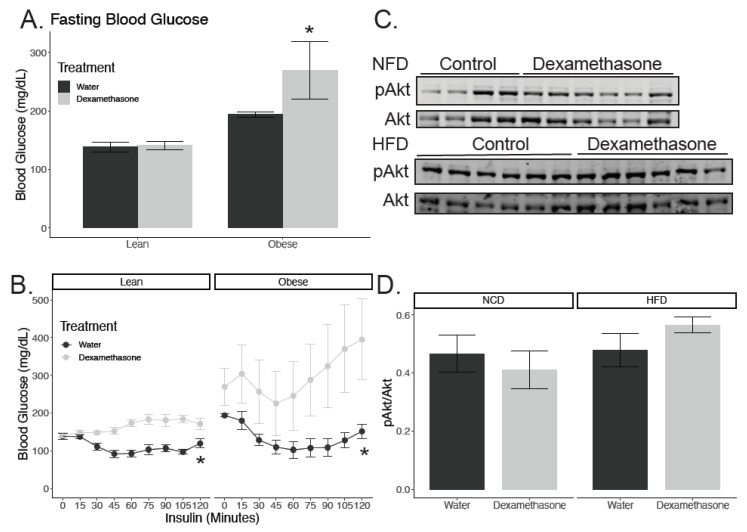
Dexamethasone treatment induces insulin resistance. Blood glucose values from lean and obese male mice after a six-hour fast and two weeks of dexamethasone or vehicle (water) treatment (**A**), followed by insulin injection (**B**). n = four mice per group. Insulin was given via intraperitoneal injection at 0.75 g/kg lean mass for lean mice and 1.5 g/kg for obese mice (n = four mice per group). (**C**) Representative western blotting of quadriceps lysates from NCD or HFD animals treated with water or dexamethasone. (**D**) Quantification of pAkt/Akt from samples described in (**C**). Asterisks indicate a significant interaction between diet and treatment by two-way ANOVA (**A**) or mixed linear models (**B**) analyzed separately for lean and obese mice.

**Table 1 biomedicines-08-00420-t001:** Body mass, fat mass, calorie and dexamethasone intake. Asterisks indicate significant interaction between diet and dexamethasone treatment by two-way ANOVA. n = 6–8 mice per group.

	NCD, Water	NCD, Dexamethasone	HFD, Water	HFD, Dexamethasone
Body weight at sacrifice (g)	31.5 ± 0.7	29.2 ± 1.5	46.5 ± 1.0	34.2 ± 1.6 *
Lean mass at sacrifice (g)	26.2 ± 0.4	23.8 ± 0.6	27.8 ± 1.1	23.7 ± 0.3
Fat mass at sacrifice (g)	3.1 ± 0.6	3.6 ± 0.5	16.0 ± 1.3	11.6 ± 1.6 *
Percent fat mass at sacrifice	9.8 ± 1.7	12 ± 1.2	34 ± 2.0	33 ± 3.2
Food intake per mouse per day during dexamethasone treatment (g)	3.5 ± 0.09	3.7 ± 0.21	2.1 ± 1.0	3.6 ± 0.31
Calorie intake per mouse per day during dexamethasone treatment (kcal)	10.1 ± 0.26	10.8 ± 0.61	9.9 ± 4.7	17.0 ± 1.5 *
Fluid intake per mouse per day during dexamethasone treatment (mL)	11.7 ± 3.0	9.3 ± 3.0	15.9 ± 1.0	8.6 ± 1.7

**Table 2 biomedicines-08-00420-t002:** Primers used in this manuscript. Key atrophy transcripts *Fbxo32* and *Trim63* (encoding Atrogin-1 and MuRF1, respectively) and their upstream regulators *Foxo1* and *Foxo3. Nr3c1* (encodes the glucocorticoid receptor) and *Pgk1* was used as a control gene.

Gene	Forward 5′-3′ Sequence	Reverse 5′-3′ Sequence
*Fbxo32*	CTTCTCGACTGCCATCCTGG	GTTCTTTTGGGCGATGCCAC
*Trim63*	GAGGGCCATTGACTTTGGGA	TTTACCCTCTGTGGTCACGC
*Foxo1*	AGTGGATGGTGAAGAGCGTG	GAAGGGACAGATTGTGGCGA
*Foxo3*	AAACGGCTCACTTTGTCCCA	ATTCTGAACGCGCATGAAGC
*Nr3c1*	CAAGGGTCTGGAGAGGACAAC	GCTGGACGGAGGAGAACTCA
*Pgk1*	CAAGCTACTGTGGCCTCTGG	CCCACAGCCTCGGCATATTT

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
