# Peer review of "Obesity Augments Glucocorticoid-Dependent Muscle Atrophy in Male C57BL/6J Mice"

_biomedicines, 2020, doi:10.3390/biomedicines8100420_

Round 1

Reviewer 1 Report

The findings are interesting, but not completely novel. Adhikary et al (Steroids 2019) published similar findings before. - Experimental design is good, with multiple readouts for muscle function/strength.  - Statistical analysis is not always clear. The figure legends explains that an arterisk means significant interaction between diet and treatment, but the graphs in Figure 1A and B both show an arterisk. The authors should clarity this.  - It is surprising that dexamethasone treatment in lean mice has so little effect on grip strength, as similar studies with dexamethasone show strong effects of dexamethasone treatment on grip strength in lean mice (Shen et al, Journal of Cachexia Sarcopenia Muslc 2019; doi. 10.1002/jcsm.12393). The authors should elucidate on the discrepancy between their study and the published one. - In several parts of the manuscript (e.g. line 104) a reference to the figure is missing. In line 145, Figure 2I should be Figure 2H. - The manuscript lacks mechanistic insight. There is no data that shows any insight on why the dexamethasone-induction of atrogenes is more robust in obese mice.  

Author Response

The findings are interesting, but not completely novel. Adhikary et al (Steroids 2019) published similar findings before. - Experimental design is good, with multiple readouts for muscle function/strength.  - Statistical analysis is not always clear. The figure legends explains that an arterisk means significant interaction between diet and treatment, but the graphs in Figure 1A and B both show an arterisk. The authors should clarity this. 

We have clarified in the legend for figure one that the asterisk does not indicate an interaction for these panels:

Asterisks indicate significant interaction between diet and treatment by two-way ANOVA except for panels A-B where it indicates a difference between treatments (n=5-8 per group).

It is surprising that dexamethasone treatment in lean mice has so little effect on grip strength, as similar studies with dexamethasone show strong effects of dexamethasone treatment on grip strength in lean mice (Shen et al, Journal of Cachexia Sarcopenia Muslc 2019; doi. 10.1002/jcsm.12393). The authors should elucidate on the discrepancy between their study and the published one

A critical differences exist between our work and that of Shen et al..  One is that they used a much higher dose of dexamethasone (25mg/kg for them, 1 mg/kg for our study).  We believe that this explains the reduced loss of grip strength in our model.  We have cited the work of Shen et al in our introduction

 In several parts of the manuscript (e.g. line 104) a reference to the figure is missing.

 In line 145, Figure 2I should be Figure 2H. –

We have updated these references, and thank the reviewer for noticing these mistakes.

The manuscript lacks mechanistic insight. There is no data that shows any insight on why the dexamethasone-induction of atrogenes is more robust in obese mice.

We agree that we have not defined the mechanism by which obesity modifies glucocorticoid actions in muscle.  We believe that the data presented in Figure 3 supports the hypothesis that obesity causes more transactivation of critical GR-dependent genes but as yet do not have a clear biochemical mechanism to why.  We are pursuing this question aggressively and look forward to identifying and sharing these answers in forthcoming work.  We speculate about a few options in the revised discussion:

One hypothesis is that obesity remodels the chromatin landscape, allowing for easier GR access to genes involved in modulating muscle size and function. Indeed, obesity alters the packing and accessibility of DNA in adipocytes [14,21,41] and therefore may have a similar effect in muscle in which Glucocorticoid Response Elements are more easily bound by GR causing increased glucocorticoid action.  Another potential mechanism is that the effects of GR-dependent signaling are enhanced by insulin resistance by FOXO dephosphorylation

Reviewer 2 Report

The manuscript by Gunder et al. describes and interesting mouse model, which combines obesity and excess of glucocorticoids. They investigate the effect of obesity and dexamethasone on several parameters including water and food consumption, body weight and fat mass, Then, they analyze several muscle features and insulin effect on blood glucose.

The manuscript is well written and well organized. The part dealing with muscle features is sound and rather exaustive. 

My only concern is about the poor description of glucose metabolism in the animal model.

In particular, they only performed an insulin tolerance test to evaluate insulin resistance. I do understand that the use of hyperinsulinemic clamp is not easy and obvious to perform, but they should at least further investigate the mechanisms of insulin resistance.

For instance, Does the lack of In vivo insulin effect relate to defects in muscle insulin action. What about insulin effect on canonical targets (muscle vs liver vs adipose tissue)? 

How the authors address the mechanism of dexa-induced defects in insulin action of obese vs lean animals

Reviewer 3 Report

Glucocorticoids widely used in clinical medicine but many side effects in skeletal muscle is  serious problem of use this hormone. The present paper is dedicated to the negative side effects of dexamethasone in skeletal muscle, effect of obesity on muscle atrophy and grip strength. The atrophy  and reduction of muscle strength mainly in type II fibers is logical as muscle fibers with low oxidative capacity are more sensitive to the catabolic effect of glucocorticoids.

This paper contain new knowledge of effect of glucocorticoids in skeletal muscle, have theoretical and practical value. Paper is well written and I suggest to publish paper without changes.

Author Response

We thank the reviewer for their comments

Reviewer 4 Report

Gunder et al. examined the effects of dexamethasone treatment on parameters of skeletal muscle atrophy in mice fed either a High-fat diet or standard chow. This work builds on their previous work by Harvey et al. (2018), where the authors observed impaired glucose tolerance, decrease fat mass, hepatic steatosis, and increased lipolysis. They demonstrate that HFD-dexamethasone animals weight less than their HFD-vechile controls despite consuming significantly more calories.  The discrepancy in the mouse body wright was due to less fat mass and lean mass. They comprehensively demonstrate that dexamethasone treatment decreases muscle strength, fibre type and cross sectional area. However, despite the reductions in muscle mass and strength, the authors did not observe differences in markers of the E3 ligases, MuRF1 and Atrogin-1.  The manuscript is well written, relevant, but could be improved from the addition of some molecular work.

Main comments

  1. Please include main effects of the diet and dexamethasone treatment either in text or present on graph, as it is hard to interpret where there are main effects.
  2. Change the title as it currently a bit misleading. I think switching “promotes” to “exacerbates” or “augments” is more suitable, as there appears to be some main effects of treatment with the dexamethasone for loss of muscle strength, CSA and mass. 
  3. Add westerns for MuRF1, Atrogin-1, FOXO3, and LC3BII/I. In its current state, the manuscript is only descriptive and would benefit from the addition of molecular explanations to the changes observed.

Other:

  1. Please include % fat free mass and lean mass at sacrifice on table 1. While this data is available to some degree on figure 2A, it would be more comprehensive and clear to also list the data at time of sacrifice on table 1 for a complete overview of body composition. Also please include the gastrocnemius weights normalized to body weight, as the reduction in mass could be attributable to the decrease in body weight in the dexamethasone treated mice.
  2. There is a formatting error on table 1 for fluid intake per day.
  3. Were mice activity levels recorded? Could changes in physical activity account for some of the differences observed?
  4. Is the fluid intake for HFD-water vs. Chow-water animals significant? Could this potential increase in fluid intake be do to impaired glucose tolerance?
  5. “In NCD animals, the force generated by nerve stimulation was reduced 10% when treated with dexamethasone.” Is this significant? As Figure 1C does not reflect this. Same for the 11% reduction for muscle force figure 1D. If not statistically significant, I think it would help to list the p values of the main effects of diet and dexamethasone for clarification.
  6. It would be better to present the muscle CSA data before presenting the muscle force-CSA regression.
  7. Please mention in text that the stain for fibre type assesses SDH activity. Is there a main effect of diet/obesity for decreased type IIa/IIb?
  8. Include the 15 day time point of gene data in figure 3 as bar graphs that show the 4 groups. Also the asterisks are missing on the current figure 3 to what is significant. From the text it looks like 7 days of treatment increases FOXO3, MuRF1 and Atrogin, but this is not reflected in the figure.
  9. It would be interesting to include western blots for MuRF1, Atrogin, phosphor and total FOXO3. Along the same lines, it would be good to include a marker of autophagic flux such as LC3II/I, as changes in autophagy could contribute to the reductions in muscle mass. 
  10. In the discussion, it is mentioned that the mechanisms contributing to selective fibre type loss following dexamethasone treatment is unclear. It would be good if the authors expanded on their current data set to include markers involved in pathways known to induce fibre type switching such as ERK1/2, MAPK etc.
  11. The primer sequence for NR3c1 is missing.

Round 2

Reviewer 2 Report

The authors have satisfactorily addressed my previous concerns.